# Improving the Targets' Trajectories Estimated by an Automotive RADAR Sensor Using Polynomial Fitting

**Georgiana Magu \*, Radu Lucaciu \* and Alexandru Isar \***

Communications Department, Politehnica University, 300223 Timisoara, Romania
\* Correspondence: georgiana.magu@upt.ro (G.M.); radu.lucaciu@upt.ro (R.L.); alexandru.isar@upt.ro (A.I.)

**Abstract:** A way to reduce the uncertainty at the output of a Kalman filter embedded into a tracker connected to an automotive RADAR sensor consists of the adaptive selection of parameters during the tracking process. Different informed strategies for automatically tuning the tracker's parameters and to jointly learn the parameters and state/output sequence using: expectation maximization; optimization approaches, including the simplex algorithm; coordinate descent; genetic algorithms; nonlinear programming using finite differencing to estimate the gradient; Bayesian optimization and reinforcement learning; automatically tuning hyper-parameters in the least squares, were already proposed. We develop here a different semi-blind post-processing approach, which is faster and more robust. Starting from the conjecture that the trajectory is polynomial in Cartesian coordinates, our method supposes to fit the data obtained at the output of the tracker to a polynomial. We highlight, by simulations, the improvement of the estimated trajectory's accuracy using the polynomial fitting for single and multiple targets. We propose a new polynomial fitting method based on wavelets in two steps: denoising and polynomial part extraction, which compares favorably with the classical polynomial fitting method. The effect of the proposed post-processing methods is visible, the accuracy of targets' trajectories estimations being hardly increased.

**Keywords:** polynomial fitting; trajectory; Kalman filter; wavelets; multiple targets

## 1. Introduction

The problem of tracking targets using the measurements of an automotive RADAR sensor supposes the integration of measurements into a longer-term picture [1]. Multiple target tracking is realized by the cooperation of two algorithms: a measurement-to-track data association algorithm and a tracks filtering algorithm. Tracks filtering, usually realized by Kalman filters, is the process of estimating the trajectory (i.e., position, velocity, and possibly acceleration) of a track from measurements (e.g., range, bearing, and elevation) that have been assigned to that track. A trajectory estimate always has an associated uncertainty, as can be observed in Figure 1. This uncertainty makes the target's localization more difficult. Hence, the reduction in this uncertainty is very important for the trajectory's accuracy and the target localization improvement. Kalman filters are widely used to estimate the state of a linear dynamical system from noisy measurements [1–4]. Despite its wide use and success, the Kalman filter is not completely automated yet, and practitioners employing the Kalman filter still must realize the manual tuning of its parameters. As a result, there is a certain incertitude at the output of the Kalman filter. A way to reduce this value consists of the adaptive selection of parameters during the filtering process. We will call such a filter an adaptive Kalman filter in the following.

Many researchers have proposed methods for automatically tuning the Kalman filter's parameters. They proposed different signal processing methods to accomplish this task. For example, they tried to jointly learn the parameters and state/output sequence. To do this, they proposed different strategies, for example, the expectation maximization algorithm, or different optimization approaches such as the simplex algorithm, the coordinate descent

algorithm, genetic algorithms, nonlinear programming using finite differencing to estimate the gradient, Bayesian optimization, reinforcement learning, and automatically tuning hyper-parameters in the least squares [2].

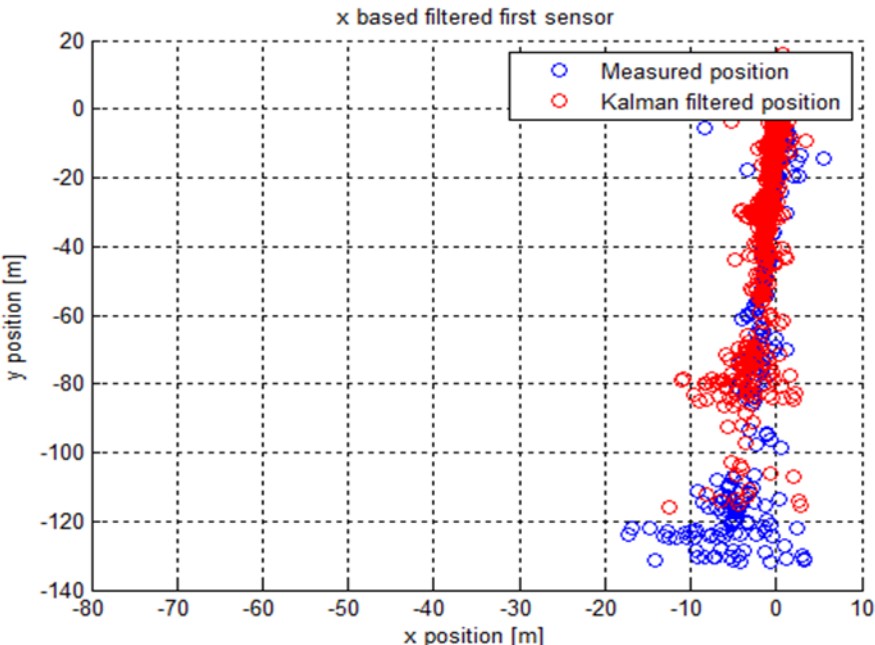

**Figure 1.** An example of a trajectory estimated by a tracker based on a Kalman filter.

In this paper, we propose a blind approach based on polynomial fitting to reduce the tracker's uncertainty. This approach starts from the conjecture that almost all trajectories encountered in automotive RADAR and global positioning system (GPS) localization problems are polynomials in rectangular coordinates because the roads are polynomials, as can be observed in the example in Figure 2.

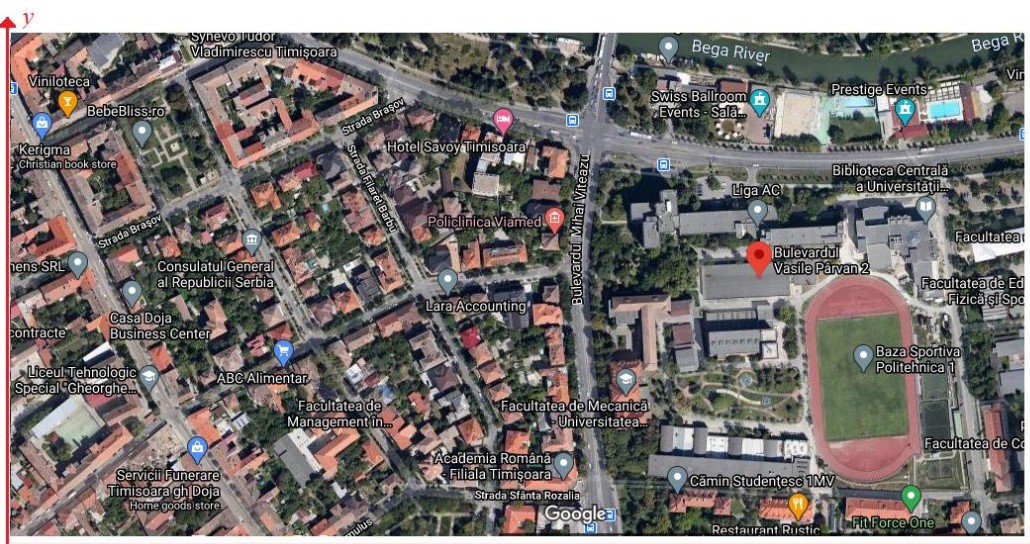

**Figure 2.** Roads in the neighborhood near our faculty building [5,6].

We highlight, by simulations, the improvement of the estimated trajectory's accuracy due to the polynomial fitting in the case of single or multiple targets, and, further to this, we propose three new polynomial fitting methods for the trajectories estimated by Kalman filters. The first two data fitting methods, which are proposed in this paper, fuse the two

trajectories, which correspond to the two coordinate axes, $x$ and $y$, obtained at the output of the Kalman filter into a polynomial $y(x)$ by separate polynomial fitting on each axis. The third proposed data fitting method uses wavelets and has two steps for each coordinate axis: denoising and polynomial part extraction. It compares favorably to the first two data fitting methods proposed. The effect of the proposed post-processing methods is visible, the accuracy of targets' trajectories estimations being hardly increased.

## 2. Related Work

One of the essential parts of the subsystems of an advanced driver assistant system (ADAS), used in automotive as collision avoidance and lane assist, is target tracking [7]. The target tracking can be realized by the cooperation of two techniques: data association and a Kalman filters bank. For each track, a separate filter is implemented. These filters compose the filters bank and operate in parallel.

Despite its wide use and success, practitioners employing the Kalman smoother still must resort to manually tuning its parameters. Due to the non-stationarity of the signals at the input of Kalman filters, uncertainties appear at their outputs.

A way to reduce this uncertainty consists of the adaptive selection of parameters during the filtering process. Many researchers have proposed methods for automatic tuning of the parameters in Kalman smoothers, based on different strategies [2]. Some of these strategies solve convex or non-convex optimization problems. One of the first methods proposed for automatically tuning Kalman's filter parameters, exploiting its ability to work with missing data, is the joint learning of parameters and state/output sequence using the expectation maximization (EM) algorithm [8]. For a given state-space model, a simple recursive procedure can be derived for the parameters' estimation by maximum likelihood, using the EM algorithm and the conventional Kalman filter algorithm in conjunction. Approaches that are more recent employ different optimization methods, including the simplex algorithm [9]. The subject of [10] is a method based on the coordinate descent algorithm for automatically learning the noise parameters of a Kalman filter. The authors of [10] propose machine-learning algorithms for automatically choosing the noise parameters of the Kalman filter or extended Kalman filter (EKF). They start from the fundamental assumption that during the EKF development, it is possible to instrument the system to measure additional variables. These new variables provide accurate estimates of the Kalman filter's state variables. These additional measurements are available only in the initial tuning phase. The authors of [11] proposed an optimal automated procedure for tuning Kalman filters based on genetic algorithm minimization, which uses statistical consistency tests. The paper [12] focuses on applying gradient-based methods to tune filters that estimate the vertical state of aircraft based on quantized altitude measurements only. Significant time and effort is frequently spent in tuning various Kalman filter's model parameters, as process noise covariance, pre-whitening filter models for non-white noise, etc. Conventional optimization techniques for tuning are non-convex optimization problems, producing poor local minima localization. To address these issues, the authors of [13] developed a new Bayesian optimization strategy for automatically tuning Kalman filters. The authors of [13] propose two stochastic objective functions: normalized estimation error squared (NEES), used when ground truth models are available, and the normalized innovation error squared (NIS), used when only sensor data are available for the optimization purpose. Bayesian optimization can efficiently identify multiple local minima and provide uncertainty quantification on its results by intelligently sampling the parameter space to both learn and exploit a "nonparametric Gaussian process surrogate function" for the NEES/NIS costs. Tuning Kalman filters is difficult and time consuming. The filter designer frequently achieves it before navigation. This process is often very expensive in terms of time and is in no way guaranteed to result in optimal parameters. Reinforcement learning is an intelligent solution to this problem, which uses a combination of dynamic programming and trial and error exploration to develop a set of optimal parameters [14]. Good estimates of the tuning parameter values could be obtained

rapidly using reinforcement learning, with significantly less iteration, in comparison to an exhaustive search. The authors of [2] propose an approach inspired by previous research on automatically tuning hyper-parameters in the least squares [15]. The least squares method is the simplest and most commonly applied computational method. The least squares objective is rarely the true objective in almost all applications. This discrepancy is accounted for in [15] by parametrizing the least squares problem and automatically adjusting these parameters using an optimization algorithm. The authors of [15] propose a method named least squares auto tuning and apply this method to data fitting. The result in [2] departs from prior work on tuning Kalman filters [8–15]. Since the Kalman filter can deal with missing measurements, some measurements can hold out and be used to evaluate it. Additionally, the method in [2] makes explicit use of the gradient of the loss concerning the Kalman filter parameters, leading to a more efficient optimization algorithm.

Another way to reduce the uncertainty at the output of the Kalman filter, which is developed in this paper, is by post processing. This method avoids the high computational complexity required by adaptive methods and speeds up the uncertainty reduction in the Kalman filter results. In [16], the wavelet-based polynomial fitting is proposed as a post processing method, analyzing a single target scene contained in real data. The polynomial fitting, as a post processing method, was applied in the case of multi target scenarios, using the Hungarian algorithm, [17], to associate measurements with targets for simulated data in [6], obtaining promising results.

## 3. Reducing the Uncertainty at the Output of Kalman Filter by Post Processing

The estimation of multiple target trajectories by an automotive RADAR sensor supposes the following operations:

- The measurement of scatters received from each target [7];
- Estimation of range, radial velocity, azimuth and elevation angles, signal to noise (SNR) and other parameters of each target;
- Target detection;
- Data association (association of the measurements with the trajectories);
- Kalman filtering of each trajectory.

The first three operations depend on the particularities of the RADAR sensor and, more specifically, on the type of waveform used. The last two operations are detailed in the following subsections, representing the methods, which were used to elaborate this paper. These two methods are applied in conjunction because the intermediate results of one of the methods serve as initializations for the other method and reciprocally.

### 3.1. Data Association

The goal of data association is to establish a correspondence between observations and existing tracks [18].

Traditionally, multiple target tracking (MTT) algorithms have been designed for scenarios with multiple remote objects situated far away from the sensor. An object is not always detected by the sensor in such scenarios, and if it is detected, it occupies, at most, one sensor resolution cell. "From traditional scenarios, specific assumptions on the mathematical model of MTT problems have evolved including the so-called "small object" assumptions" [19]:

- The evolution of objects is independent;
- The model of each object is a point at a given time;
- Each object produces at most a single measurement per scan/time frame.

Due to: sensor noise, missed detections, clutter detections, measurement origin uncertainty and an unknown and time-varying number of targets, MTT based on the "small object" assumptions is a highly complex problem. Theoretically, each measurement can be associated with a different track. For an increased number of measurements, a very high number of associations is possible. To find the true data association, all these possible

associations would need to be verified. This requires a tremendous volume of computation. To reduce this volume of computation, at the beginning of the data association procedure, a validation region (gate) is formed around the predicted track position, to eliminate unlikely observation-to-track pairing. The measurements observed in the interior of this gate are good candidates for the association with the considered track. An observation can be associated with a track and used to update the track filter if it is the single observation in the interior of the corresponding gate and if it is not in the interior of the gate of any other track. When an observation falls within the gates of multiple target tracks or when multiple observations fall within the gate of a target track in a dense target environment, additional logic is necessary.

For multiple sensors and multiple targets, common association algorithms are:

- Joint likelihood (JL): a variation of the multi-hypothesis track splitting algorithm extended to multiple tracks [20];
- Joint probabilistic data association (JPDA): for each track, this algorithm updates the filter based on a joint probability of association of the latest set of measurements and each track [19];
- Multiple hypothesis joint probabilistic (MHJP): a variation of the optimal Bayesian filter using joint probabilities among multiple track associations for multiple hypotheses [18];
- Random finite sets (RFS) approaches: rely on modelling the objects and the measurements as random sets.

Inspired by [6,21], we use in this paper the Hungarian algorithm (Kuhn–Munkres algorithm) [17] for data association and the Kalman filter for tracking and realizing the tracking of multiple radar targets. The Hungarian algorithm deals with the problem of personnel assignment. It aims to find the best assignment of a set of persons to a set of jobs. In RADAR tracking, the necessity to correctly assign the position of a detected object to a track can be completed by computing a "distance" matrix between the track estimates at the current timestamp and the measurements. The Hungarian algorithm performs the optimal assignment between the tracks (rows of the matrix) to the measurements (columns). The result is a column vector that contains the assigned column number (measurement) in each row (track). The absence of measurements assigned at the current step for the respective track is marked with a zero value in this matrix.

### 3.2. Kalman Filter

One of the original applications of the Kalman filter [3] was for space aircraft tracking in the Apollo navigation system. The Kalman filtering is used in a tracking process to realize a system state estimation at the current timestamp $t$ [1,7,16]. This operation is accomplished by combining the values of the measurements at the current timestamp $t$ with the system state prediction realized at the previous time stamp $t-1$. The solution is computed recursively by the filter. The current system state at timestamp $t$ consists of the Cartesian coordinates of position and speed:

$$\mathbf{x} = \begin{bmatrix} x & y & v_x & v_y \end{bmatrix}^T \tag{1}$$

where $T$ denotes the transpose. Considering the simplest model for the target movement (constant velocity), the state prediction for timestamp $t$ based on the previous state can be described as:

$$\mathbf{x}_t = \mathbf{A} \cdot \mathbf{x}_{t-1} + \mathbf{w}_{t-1} \tag{2}$$

where $\mathbf{A}$ is the state transition matrix and $\mathbf{w}_{t-1}$ is Gaussian state noise vector of zero expectation and covariance matrix $\mathbf{E}_x$, or, equivalently:

$$
\begin{bmatrix} x \\ y \\ v_x \\ v_y \end{bmatrix}_t = \begin{bmatrix} 1 & 0 & T & 0 \\ 0 & 1 & 0 & T \\ 0 & 0 & 1 & 0 \\ 0 & 0 & 0 & 1 \end{bmatrix} \begin{bmatrix} x \\ y \\ v_x \\ v_y \end{bmatrix}_{t-1} + \begin{bmatrix} 0 \\ 0 \\ w_x \\ w_y \end{bmatrix}_{t-1}
\tag{3}
$$

where T is the measurement period. The measurement equation is written as:

$$
\mathbf{z}_t = \mathbf{H} \cdot \mathbf{x}_t + \mathbf{n}_t
\tag{4}
$$

where $\mathbf{H}$ is the measurement matrix and $\mathbf{n}$ is the measurement noise that is the supposed Gaussian of zero mean and covariance matrix $\mathbf{E}_z$. When only the target positions are estimated (as we will proceed in the following), the measurement matrix is expressed as:

$$
\mathbf{H} = \begin{bmatrix} 1 & 0 & 0 & 0 \\ 0 & 1 & 0 & 0 \end{bmatrix}
\tag{5}
$$

The Kalman filtering equations describe the current state prediction and the update of the state estimation as [7,16]:

$$
\begin{aligned}
\bar{\mathbf{x}}_t &= \mathbf{A} \cdot \mathbf{x}_{t-1} \\
\bar{\mathbf{P}}_t &= \mathbf{A} \cdot \mathbf{P}_{t-1} \cdot \mathbf{A}^T + \mathbf{E}_x \\
\mathbf{K}_t &= \bar{\mathbf{P}}_t \cdot \mathbf{H}^T \cdot \left( \mathbf{H} \cdot \bar{\mathbf{P}}_t \cdot \mathbf{H}^T + \mathbf{E}_z \right)^{-1} \\
\mathbf{x}_t &= \bar{\mathbf{x}}_t + \mathbf{K}_t \cdot \left( \mathbf{z}_t - \mathbf{H} \cdot \bar{\mathbf{x}}_t \right) \\
\mathbf{P}_t &= \left( \mathbf{I} - \mathbf{K}_t \cdot \mathbf{H} \right) \cdot \bar{\mathbf{P}}_t
\end{aligned}
\tag{6}
$$

where $\mathbf{K}$ denotes the Kalman gain, $\bar{\mathbf{P}}$ is the predicted covariance matrix of the error and $\mathbf{P}$ is the covariance matrix of the state estimation error. This error expresses the output of the Kalman filter uncertainty mentioned earlier. In the traditional formulation, the dynamics and output matrices are considered attributes of the system, the covariance matrices of the process and sensor noise are tuned by the designer, within some limits, to obtain good performance in simulation or on the actual system [2,16]. For the rest of this paper, the covariance matrices for the measurement noise $\mathbf{E}_z$ and the state error estimation $\mathbf{E}_x$ are selected based on the following criteria [16]:

- Small values for the elements of the matrix $\mathbf{E}_z$ mean that we tend to rely more on the noisy measurement than on the prediction of the filter based on the model;
- Big values for the elements of the matrix $\mathbf{E}_x$ mean that we do not rely very much on the model.

The measurement noise covariance matrix $\mathbf{E}_z$ and the state error estimation covariance matrix $\mathbf{E}_x$ are [6]:

$$
\mathbf{E}_z = \begin{bmatrix} z_x & 0 \\ 0 & z_y \end{bmatrix}
\tag{7}
$$

and

$$
\mathbf{E}_x = \begin{bmatrix} \frac{T^4}{4} & 0 & \frac{T^3}{2} & 0 \\ 0 & \frac{T^4}{4} & 0 & \frac{T^3}{2} \\ \frac{T^3}{2} & 0 & T^2 & 0 \\ 0 & \frac{T^3}{2} & 0 & T^2 \end{bmatrix}
\tag{8}
$$

As already mentioned, the Kalman filters bank cooperates with the data association procedure during the target tracking. At a fixed moment, the Kalman filter, which tracks a specified target, is informed by the data association algorithm what the measurements are that he can use to predict the following state of the system. After the accomplishment of the prediction, the Kalman filter informs the data association algorithm about the new

trajectory. Then, the data association algorithm searches the new measurements for the ones that are the most appropriate for these new trajectories. Hence, the collaboration between the data association algorithm and the Kalman filter bank is an iterative process as well.

### 3.3. Reducing the Uncertainty at the Output of Kalman Filter

By analyzing Equations (7) and (8), we can understand why, despite its wide use and success, practitioners using the Kalman filter must resort to manually tuning its parameters. As a result of noise presence, possible missing data, and imperfections of models, at the output of the Kalman filter, a certain uncertainty appears. As already mentioned, this uncertainty can be reduced by transforming the Kalman filter into an adaptive filter or by post processing. A blind post processing approach, which is faster than the adaptive approaches already mentioned, because it is not iterative, and more robust because the blind estimation approaches are more robust than the informed estimation approaches, based on data fitting, was proposed in [16] and developed in [6]. This method supposes to fit the data obtained at the output of the Kalman filter to a polynomial.

#### 3.3.1. Polynomial Fitting

The meaning of data fitting is the finding of a mathematical description for data [22]. This mathematical description should be as precise as possible. A measure for the deviation of the model with data, commonly used in practice, is the root mean squared error (RMSE). So, fit means to find a minimum of the RMSE. To achieve this, a fit function f, which generally contains adjustable parameters, the fit parameters: $a_i$, $i = 0 \ldots n$, must be defined. Let us consider the simple case of a set of data $y_t$, $t = 1, \ldots, M$ that are collected as a function of one independent variable x at the points $x_t$. We would like the fit function f to describe these data in an approximate way as:

$$y_t \approx f(a_0, \ldots, a_n, x_t) \tag{9}$$

To find the parameters $a_i$ for the best approximation, the sum over the squared residuals must be minimized:

$$r = \sum_t (y_t - f(a_0, \ldots, a_n, x_t))^2 \tag{10}$$

Forcing the derivatives of r concerning all $a_i$ to be zero, we can find the minimum of RMSE:

Therefore, data fitting is a convex optimization problem. Polynomial fitting supposes to use the simplest type of fit functions: polynomials,

$$\frac{\partial r}{\partial a_0} = 0, \ \frac{\partial r}{\partial a_1} = 0, \ \ldots, \frac{\partial r}{\partial a_n} = 0. \tag{11}$$

$$f(a_0, \ldots, a_n, x_t) = a_0 + a_1 x_t + \ldots + a_n x_t^n \tag{12}$$

In the case of the polynomial fitting, Equation (11) becomes a system of $n + 1$ linear equations for the $n + 1$ unknowns $a_i$. The system may be solved by standard methods for linear equations. Generally, no approximate optimization procedures are necessary because the system of equations has one unique solution. The disadvantage of this data fitting method is the necessity to a priori know the degree of the polynomial $f(a_0, \ldots, a_n, x_t)$, $n$.

In the case of data at the output of Kalman filter, we have two sequences of coordinates: $\{x_t\}$ and $\{y_t\}$ (as it can be observed in Equations (1) or (3)), which represent samples of the horizontal and vertical axes of the reference system where the trajectory of target's movement is represented. To realize the polynomial fitting of those data, we can proceed in two steps: first, we apply the fitting procedure to the data $\{x_t\}$ and, next, we apply the fitting procedure to the data $\{y_t\}$, because by composing two polynomial functions, another polynomial function is obtained. This method will be named in the following

implicit polynomial fitting. The result is polynomial, but the chronology is not necessarily kept. Another disadvantage is the possible loss of the extrema of one or both data sequences $\{x_t\}$ and/or $\{y_t\}$ as a consequence of the smoothing effect of the implicit polynomial fitting method. This disadvantage can be counteracted by applying explicit polynomial fitting. This polynomial fitting method has the following steps:

- Polynomial fitting of the sequence $\{x_t\}$: $x = P(t)$;
- Inverting the result obtained: $t = P^{-1}(x)$;
- Polynomial fitting of the sequence $\{y_t\}$: $y = Q(t)$;
- Substitution of the result of the second step into the result of the third step: $y = Q\big(P^{-1}(x)\big)$.

Some precautions must be taken for the application of the explicit polynomial fitting method regarding the monotony of functions $P$ and $Q$ to obtain a correct result because the polynomials are invertible functions only on intervals. Another disadvantage of the polynomial fitting method is the high computation complexity required in the case of great polynomial degrees. When n has a big value, the procedure of solving the system in Equation (11) implies a high computation complexity. The procedure of solving the system of Equation (11) can be avoided by directly extracting the polynomial parts of the sequences $\{x_t\}$ and $\{y_t\}$. The direct extraction of the polynomial part of a signal can be performed in the wavelets' domain.

### 3.3.2. Wavelet-Based Polynomial Fitting

Wavelets are defined starting from an orthogonal multiresolution analysis. Adapting the signal resolution allows one to process only the relevant details for a particular task. For example, in computer vision, it is possible to process a low-resolution image first and then selectively increase the resolution when necessary. The mathematical formalization of the multiresolution concept in the space of finite energy analogic signals ($\mathbf{L}^2(\mathbf{R})$) is made in [23] in definition 7.1 as a sequence of closed subspaces $\{\mathbf{V}_j\}_{j\in\mathbf{Z}}$ satisfying six properties. Denoting the orthogonal complement of $\mathbf{V}_j$ in $\mathbf{V}_{j-1}$ by $\mathbf{W}_j$, it is shown in [23] that the sequence of closed subspaces $\{\mathbf{W}_j\}_{j\in\mathbf{Z}}$ constitutes an orthogonal decomposition of $\mathbf{L}^2(\mathbf{R})$. The subspaces $\mathbf{V}_j$ are generated by orthonormal bases $\left\{\varphi_{j,k}\right\}_{k\in\mathbf{Z}}$ formed by scaling and translating with integers a function $\varphi(\tau)$ named the scaling function (or father wavelets), $\varphi_{j,k}(\tau) = 2^{-\frac{j}{2}}\varphi\big(2^{-j}(\tau - k)\big)$. Similarly, the subspaces $\mathbf{W}_j$ are generated by orthonormal bases formed by scaling and translating with integers a function $\psi(\tau)$ named mother wavelets (MW), $\psi_{j,k}(\tau) = 2^{-\frac{j}{2}}\psi\big(2^{-j}(\tau - k)\big)$ [23]. One of the most interesting properties of a MW, which characterizes its regularity, is its number of vanishing moments (VM). A MW has r vanishing moments if:

$$\int_{-\infty}^{\infty} \tau^k \psi(t)d\tau = 0, \text{ for } 0 \le k < r \tag{13}$$

Any finite energy signal, $x(\tau)$, can be decomposed into a wavelet series:

$$x(\tau) = \sum_{k=-\infty}^{\infty} a_{j_0,k}\varphi_{j_0,k}(\tau) + \sum_{j=j_0}^{\infty} d_{j,k}\psi_{j,k}(\tau) \tag{14}$$

where:$a_{j,k} = \left\langle x(\tau), \varphi_{j,k}(\tau) \right\rangle$ are named approximation coefficients and $d_{j,k} = \left\langle x(\tau), \psi_{j,k}(\tau) \right\rangle$ are named detail coefficients. These sequences of coefficients represent the outputs of the discrete wavelet transform (DWT) of the signal x. The connection between the wavelets theory and polynomials is given by the following remark: a wavelet with r VM is orthogonal to polynomials of degree r-1 [23] (p. 166). So, if the signal $x(\tau)$ is a polynomial, all the detail coefficients $d_{j,k}$ in the right-hand side of Equation (14) equal zero. The representation

of polynomials in wavelet series is sparse (only the first sum in the right-hand side of Equation (14) is not null). The wavelet-based polynomial part extraction of a signal consists of the computation of its DWT followed by the retention of approximation coefficients only. Next, the inverse DWT (IDWT) is computed. As the sequences $\{x_t\}$ and $\{y_t\}$ at the output of the Kalman filter are affected by the already mentioned uncertainty, the mathematical model of these sequences is a sum of the polynomial part and a noise. For this reason, it is recommended to apply a denoising procedure before the polynomial part extraction. The denoising operation is another practical application of wavelets theory. We will apply in the following the denoising method proposed by Donoho [24]. It consists of three steps:

- Computation of DWT and separation of detail coefficients;
- Nonlinear filtering of detail coefficients using (for example) the hard thresholding filter [24];
- Concatenation of the sequence of approximation coefficients, extracted in the first step with the new sequence of detail coefficients obtained after the second step and the computation of the IDWT.

As in the case of the polynomial fitting methods presented in the previous section, the principal inconvenience of the proposed wavelet-based polynomial fitting method is the necessity to know a priori the degree of the fitting polynomial, r-1, to deduce r—the regularity of the MW that must be used for polynomial part extraction. This inconvenience could be eliminated using local polynomial approximations (LPA) [25].

## 4. Results

We have considered two types of data for the experimental part of this paper: simulated data and real data. We have generated the simulated data in Matlab. We obtained the real data in the framework of collaboration with Hella Timisoara. There are two types of real data: video generated by two cameras mounted on the car carrying the RADAR sensor and RADAR data generated by two receive antennas belonging to the RADAR sensor. The video data are in the format of a movie, presenting the evolution of a single target. The 24 GHz RADAR sensor used has a cycle count of 50 ms and a maximal detection distance of 105 m. Taking into account the fact that the sensor generates 336 different signals for each receive antenna, we have used a parser, which extracts the signals necessary to characterize the moving of the target. Between these signals, there are signals necessary for tracking, for example, the signals: cycle count of measure, range, speed, and angle. We have applied some signal pre-processing operations, such as separating the moving targets of clutter, conversion of polar to Cartesian coordinates, geometric-based data association [16], obtaining the sequences $\{x_t\}$ and $\{y_t\}$, which represent the input data for the last experiment reported in the following section.

We present the following two types of results: for simulated data and real data. In the case of simulated data, we evaluate the accuracy of the proposed methods by comparing the trajectories generated by simulation (input data) with the trajectories estimated by these methods (output data).

### 4.1. Simulated Data

The goal of this subsection is to test the tracking based on data association using the Hungarian algorithm and the Kalman filtering and to evaluate the reduction in uncertainty at the output of Kalman filters based on the explicit polynomial fitting method or wavelet-based polynomial fitting method. We designed two categories of experiments to accomplish this goal. The first category of experiments is designed for the evaluation of the reduction in uncertainty at the output of Kalman filter using the explicit polynomial fitting method in the case of multiple targets with linear trajectories. The goal of the second category of experiments is the evaluation of the reduction in uncertainty using the wavelet-based polynomial fitting in the case of a single target with nonlinear trajectory.

For the first step of the experiments, we designed three simulation scenarios, in each one, three targets were considered. For the simulation of a target, we generated two first

degree polynomial trajectories $x[t]$ and $y[t]$, one for each geometrical coordinate, $x$ and $y$, defining initial points and considering that the targets' movement on each geometrical coordinate is uniform with a defined velocity [21]. Velocity profiles are also generated for each geometrical coordinate $v_x[t]$ and $v_y[t]$. We have registered the analytical expressions of the trajectories and velocities for each geometrical coordinate obtained for further evaluation of the quality of the reduction in uncertainty realized by the explicit polynomial fitting method. To simulate the measurements' uncertainty, an additive white Gaussian noise (AWGN) was added to each signal: $x[t]$, $y[t]$, $v_x[t]$ and $v_y[t]$, obtaining the signals $x_p[t]$, $y_p[t]$, $v_{x_p}[t]$ and $v_{y_p}[t]$, $p = 1, 2, 3$, with a signal to noise ratio (SNR) of 0 dB. After the generation of three trajectories for each scenario, we generated the targets as well, by inserting "detections" in each point of each trajectory. Finally, we generated the input data for each experiment, combining the parameters of the three targets: trajectories and velocity profiles $x_p[t], y_p[t]$, $v_{x_p}[t]$ and $v_{y_p}[t], p = 1, 2, 3$. The second step of the experiments realized with simulated data consists of the tracking simulation, based on the cooperation of the Hungarian algorithm with the Kalman filter algorithm. The final step of each experiment consists of the application of the explicit polynomial fitting method for the reduction in the uncertainty at the output of the tracker.

4.1.1. First Experiment

In the first scenario (Figure 3), the first two targets move vertically from bottom to top and the third target moves vertically from top to bottom. The initial coordinates of the targets are target one: x = −3 m, y = −140 m, target two: x = 0 m, y = −140 m and target three: x = 3 m, y = 0 m. The positions of the three targets at different moments are shown in Figure 3a. Applying the Hungarian data association algorithm in conjunction with the Kalman filtering, the trajectories, in Figure 3b, are obtained. After applying the explicit polynomial fitting procedure to the data corresponding to each target, we obtain the results represented in Figure 3c. The estimated trajectories are identical to the initial trajectories.

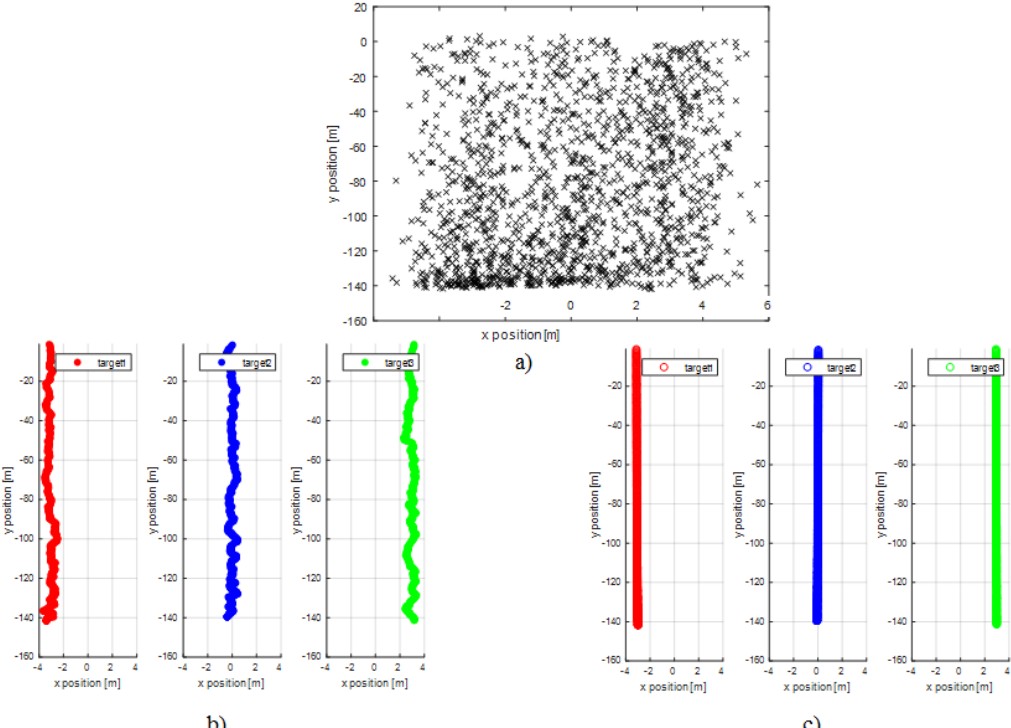

**Figure 3.** Partial and final results of the first experiment with simulated data, [6]: (**a**) spatial spreading of targets; (**b**) the cumulative effect of data association and Kalman filtering; (**c**) the effect of the explicit polynomial fitting method.

### 4.1.2. Second Experiment

In the second scenario (Figure 4), the first target moves horizontally from left to right, the second target moves vertically from bottom to top and the third target moves vertically from top to bottom. This time, the initial coordinates of the targets are target one: x = −80 m, y = −70 m, target two: x = 0 m, y = −140 m and target three: x = 3 m, y = 0 m. The measurements corresponding to the three targets, selected by the Hungarian data association algorithm, are shown in Figure 4a. The trajectories of the three targets generated by Kalman filters associated with each target can be seen in Figure 4b. After the application of the explicit polynomial fitting procedure, the trajectories represented together in Figure 4c and separately in Figure 4d are obtained.

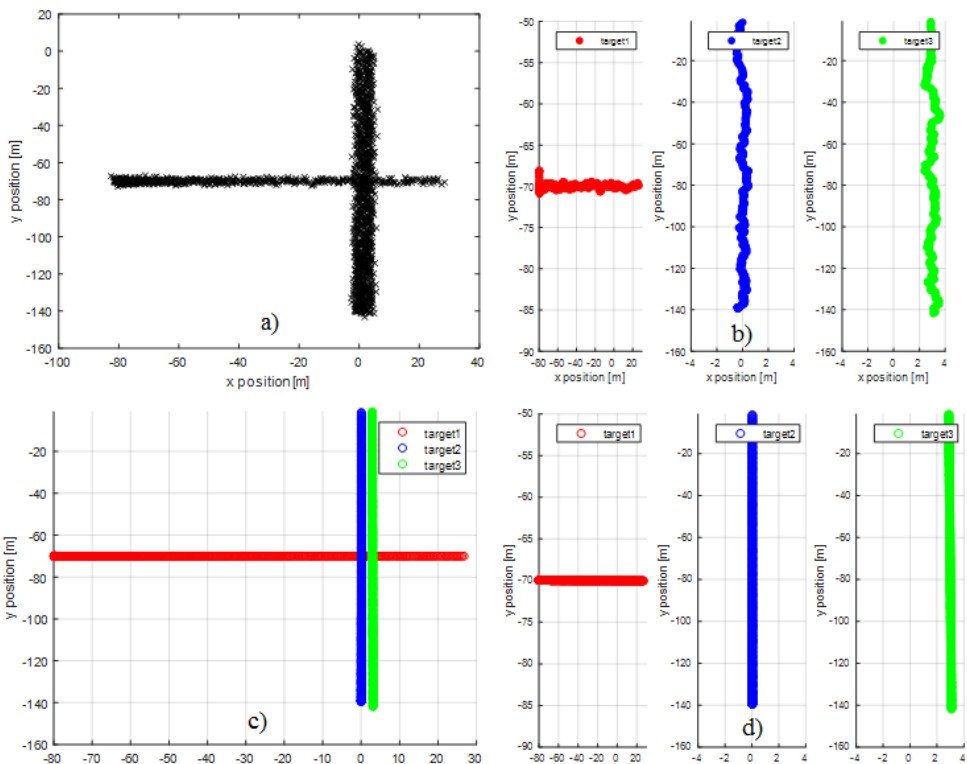

**Figure 4.** Partial and final results of the second experiment with simulated data, [6]: (**a**) spatial spreading of targets after data association; (**b**) the effect of Kalman filtering; (**c,d**) the effect of the explicit polynomial fitting method.

The estimated trajectories are identical to the initial trajectories [6].

### 4.1.3. Third Experiment

In the third scenario (Figure 5), the first target moves horizontally from left to right, the second target moves vertically from bottom to top and the third target moves obliquely from bottom to top. The initial coordinates of the targets are target one: x = −80 m, y = −70 m, target two: x = 0 m, y = −140 m, target three: x = −65 m, y = −130 m. The measurements corresponding to the three targets, shown in Figure 5a, were obtained by applying the Hungarian data association algorithm. After applying Kalman filtering to each target, the trajectories in Figure 5b were obtained. The trajectories represented in Figure 5c are obtained after the application of the explicit polynomial fitting procedure.

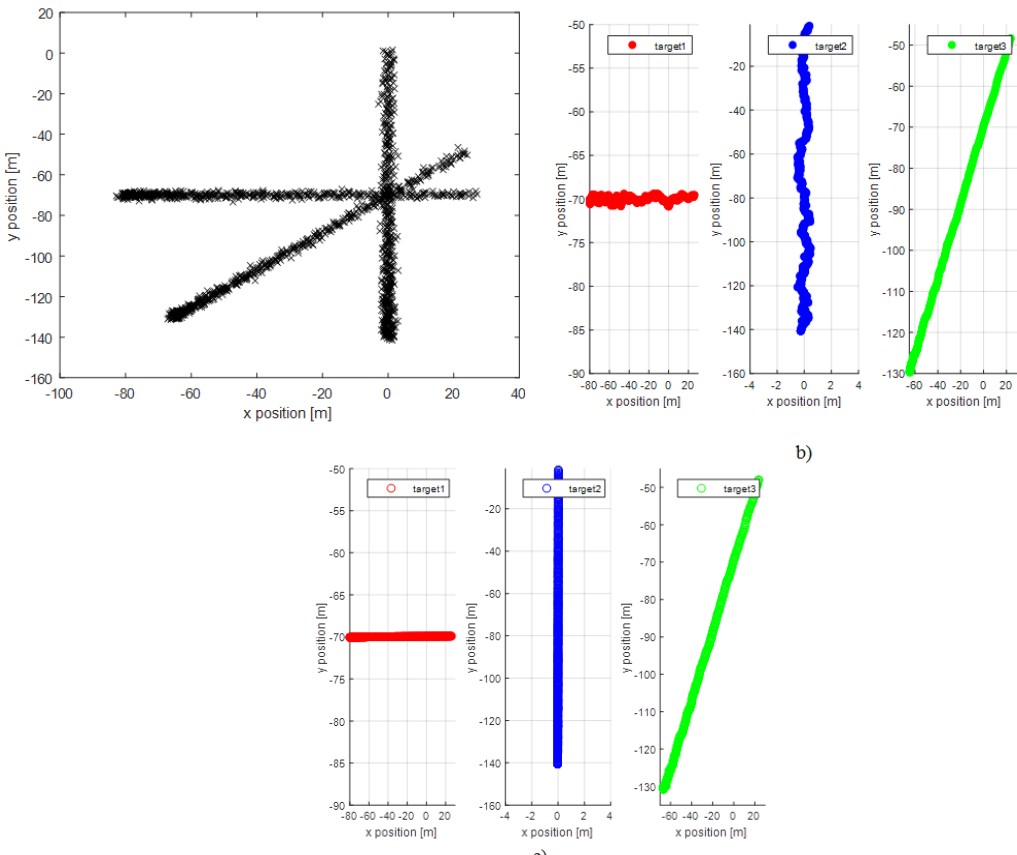

**Figure 5.** Partial and final results of the third experiment with simulated data: (**a**) spatial spreading of targets after data association; (**b**) the effect of Kalman filtering; (**c**) the effect of the explicit polynomial fitting method.

These first three experiments prove that the explicit polynomial fitting procedure is efficient in the case of first order polynomial trajectories. We have compared the useful parts of the nine trajectories generated by simulation (three trajectories in each experiment), registered in the simulation phase, with the corresponding trajectories obtained after the application of the explicit polynomial fitting procedure proposed in this paper and we have remarked that those trajectories are identical.

Hence, in the case of linear trajectories, the explicit polynomial fitting procedure can remove the entire uncertainty obtained at the output of Kalman filters.

### 4.1.4. Fourth Experiment

The fourth experiment belongs to the second category of experiments. We have realized this experiment for the evaluation of the potential of the wavelet-based polynomial fitting method. We considered the case of a single target, but this time we generated a third order polynomial trajectory, shown in Figure 6a. We have added AWGN on the component corresponding to $y$ axis obtaining a signal with a SNR of 0 dB. After the simulation of the tracking based on the cooperation of the Hungarian data association algorithm and the Kalman filter algorithm, we obtained the simulated trajectory in Figure 6b. Next, we applied the wavelet-based polynomial fitting method proposed in this paper, obtaining the result in Figure 6c. We have denoised the sequence $\{y_t\}$ using the Daubechies MW with four vanishing moments (Daubechies eight). The threshold for the hard thresholding filter was selected following the rule of 3σ. We have estimated the standard deviation of the noise component that remained in the signal in Figure 6b, σ, and we have selected the value 3σ for the threshold of the hard thresholding filter used for denoising. Next, we have extracted the polynomial part of the result obtained, computing once again the DWT, using

the same MW because it has four vanishing moments, as it is required for a polynomial of third-degree extraction, retaining only the approximation coefficients and computing the IDWT of the result (using the same MW).

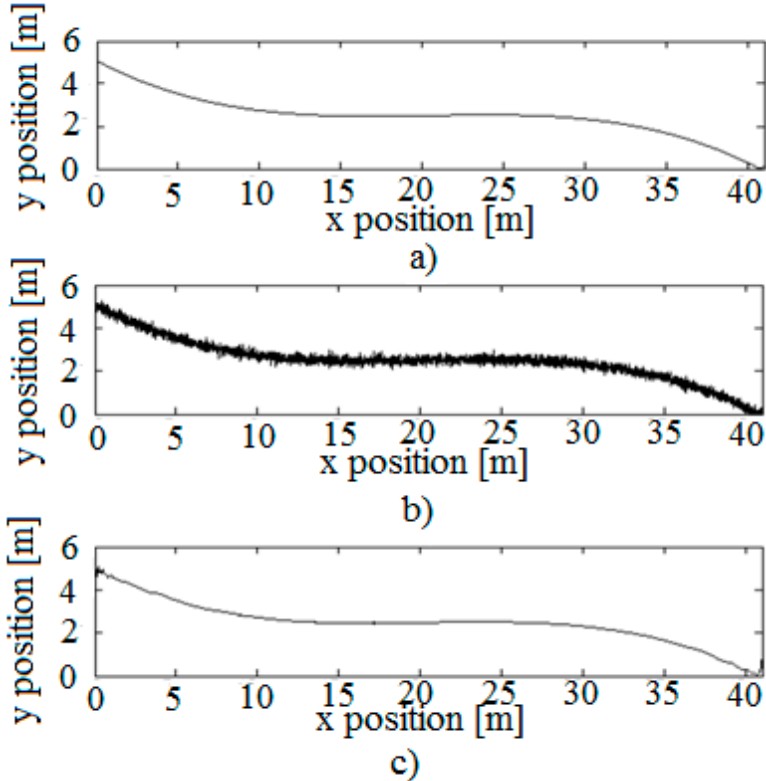

**Figure 6.** Results of the fourth experiment with simulated data: (**a**) third order polynomial trajectory; (**b**) waveform of the result obtained after the simulated tracking; (**c**) the effect of the wavelet-based polynomial fitting method.

Comparing the waveforms at the top and bottom of Figure 6, it can be said that only the border errors affect the estimation precision. These border errors are a consequence of the DWT initialization problem.

Therefore, the wavelet-based polynomial fitting procedure is effective even in the case of third order polynomial trajectories, practically removing the entire uncertainty from the output of the Kalman filter.

### 4.2. Real Data

The two types of data: video generated by two cameras, and RADAR generated by an automotive sensor with two receive antennas, were obtained from an experiment concerning the moving of a car. In Figure 7, some frames extracted from the video data are presented.

The scene recorded by the cameras was registered by a 24 GHz RADAR sensor with two receive antennas as well. The specific parameters of this RADAR sensor are the following:

- Tx modulation scheme—rapid chirps;
- Operation bandwidth—200 [MHz];
- Field of view (10 dBsm)—$0° \pm 75°$ @ 20 [m];
- Detection range (FoV):
  - 5 dBsm @ 0° azimuth—68 [m]
  - 10 dBsm @ 0° azimuth—105 [m]
- Range resolution—0.75 [m];

- Range precision—0.3 [m];
- Speed resolution—0.24 [m/s];
- Speed precision—0.05 [m/s];
- Azimuth angle precision—0.4 [°];
- Cycle count—50 [ms].

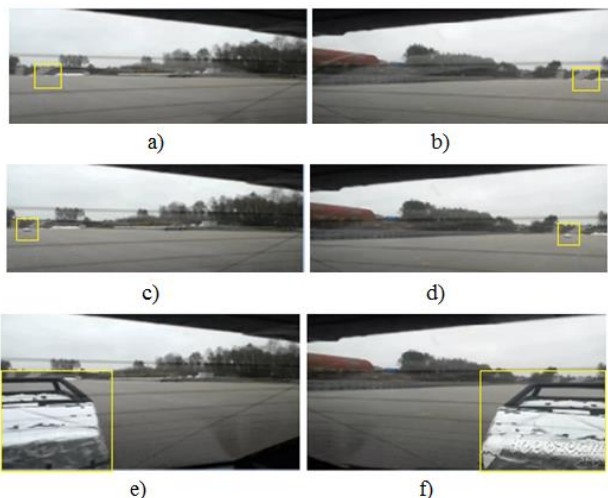

**Figure 7.** Real video data used for the last experiment, [16]: (**a**) right camera before car starting; (**b**) left camera before car starting; (**c**) right camera during car moving; (**d**) left camera during car moving; (**e**) right camera after car stopping; (**f**) left camera after car stopping.

We pre-processed the corresponding RADAR signals: separating the moving targets of clutter, converting the polar coordinates of the target into Cartesian coordinates, and applying a geometric-based data association of the moving targets' measurements. Next, we applied Kalman filtering to the obtained data. The blue points in the example in Figure 1 represent the result of geometrical data association of measurements delivered by one of the receive antenna and the red points represent the result of Kalman filtering of those data. Next, we achieved the fusion of results obtaining the blue points in Figure 8. The bottom zone, in this figure (marked in yellow), contains untrusting target positions whose distances towards sensors exceed the maximal detection distance. The middle zone in Figure 8, marked in orange, contains target positions recorded before the car starts, corresponding to the frames in Figure 7a,b. The upper zone in Figure 8, marked in light blue, corresponds to the frames of video recording neighbors of the frames in Figure 7e,f. These frames show the target car stopping maneuver. Therefore, the target car's movement starts at a distance of approximately 85 m, follows a linear trajectory and stops near the stationary car carrying the sensor. The spreading of the blue points illustrates the uncertainty at the output of Kalman filter. To reduce this uncertainty, we applied the implicit polynomial fitting procedure and the wavelet-based fitting procedure. We first applied the implicit polynomial fitting procedure, imposing a polynomial degree equal to one to the sequences $\{x_t\}$ and $\{y_t\}$, obtaining the coordinates of the points represented in green in Figure 8. So, the green line in Figure 8 represents the result of the implicit polynomial fitting procedure.

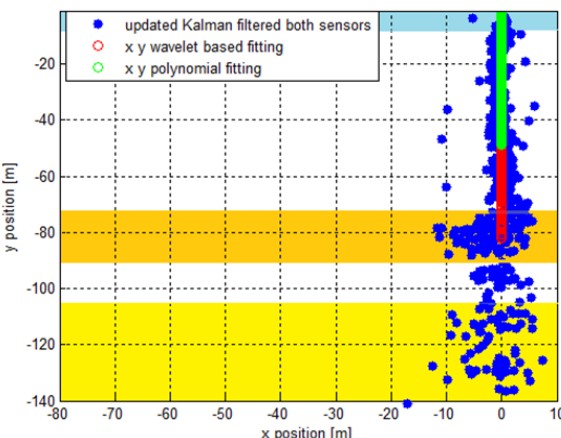

**Figure 8.** The results of the last experiment [16].

The uncertainty at the output of the Kalman filter was eliminated for the car positions situated at a distance towards the RADAR sensor inferior to 50 m.

Next, we applied the wavelet-based polynomial fitting method. For the sequence $\{x_t\}$, we applied Donoho's denoising method, computing the DWT with the aid of Daubechies 20 MW and six levels of decomposition. We extracted the polynomial part of the result. We computed the DWT using the Haar MW (which has a single VM) with eight decomposition levels obtaining the horizontal coordinates of the points marked in red in Figure 8. We modelled the sequence $\{x_t\}$ with a zero-degree polynomial. For the denoising of the sequence $\{y_t\}$, we used the Daubechies 10 MW to compute the DWT with six decomposition levels and for the extraction of the polynomial part of the result we used the Daubechies four MW (which has two VM) to compute a DWT with eight decomposition levels. After the elimination of the detail wavelet coefficients and the computation of the IDWT, we obtained the vertical coordinates of the points belonging to the red line in Figure 8. We supposed that this sequence can be modeled by a first-degree polynomial. The result of the wavelet-based polynomial fitting method is a vertical line between a point situated at approximately 90 m from the sensor and the neighborhood of the sensor. A part of the red line in Figure 8 is covered by the green line. So, the wavelet-based polynomial fitting method is better than the implicit polynomial fitting method, because it estimates the entire trajectory of the car, starting in the orange zone and finishing in the light blue zone. The trajectory estimated with the aid of the implicit polynomial fitting method is incomplete. The wavelet-based polynomial fitting method is efficient, being accurate and keeping the integrity of the trajectory.

## 5. Discussion

Despite the fact that research in the past few years has introduced many new estimation algorithms, the Kalman filter remains one of the most widely used algorithms today, being efficient, simple and robust. However, due to different reasons, for example, system and/or noise modeling imperfections (see Equations (2)–(4), (7) and (8)), missing data, measurement errors, discretization in time, insufficient convergence speed, a certain incertitude appears at the output of the Kalman filter.

The goal of this paper was to reduce and even to eliminate this incertitude. There are two strategies to accomplish this goal: a solution proposed in literature based on the adaption of the Kalman filter by continuously updating its parameters and the solution proposed in this paper based on post-processing the results of the Kalman filter by data fitting algorithms.

Let us consider first the adaptive Kalman filter solution. The adaptive approach in [2] is a form of Kalman's filter quadratic optimization problem and realizes the automatic tuning of the filter's hyper-parameters in least squares. The method in [2] makes explicit use of the gradient of the loss with respect to the Kalman filter's parameters, leading to an

efficient optimization algorithm. In [10], the authors propose a method for automatically learning the Kalman filter's noise parameters. The goal of [11] is an automated procedure for tuning Kalman filters based on a genetic algorithm. The authors of [12] regard the tuning of the Kalman filter as a convex optimization problem and solve the problem by applying gradient-based methods. The Bayesian non-convex optimization strategy for automatically tuning Kalman filters proposed in [12] identifies multiple local minima. The authors of [14] present a method to reduce the uncertainty at the output of the Kalman filter. They use this method for global positioning system (GPS) localization by cooperation with micro-electro-mechanical sensors (MEMS) in the framework of an integrated navigation system (INS). When the GPS signals are blocked or lost due to multipaths, the target's tracking can be continued using the signals delivered by the MEMS, since these sensors are fully self-contained and do not rely upon external signals. Furthermore, the absolute positions from GPS can filter and minimize the cumulative drift of the dead reckoning inertial sensors. This creates a continuous navigation solution with the accuracy determined by the quality of both the GPS signals and inertial sensors used. In this case, the Kalman filter combines measurements from GPS with those of MEMS using certain a priori information. The filter requires knowledge of the system and measurement dynamics as well as a statistical description of the system noises, measurement errors and uncertainty in the dynamic model. This includes the noise characteristics of both the MEMS and GPS updates. The filter then takes several assumptions, such as white noise behavior and Gauss–Markov properties, to weight the measurements in an optimal manner. Reinforcement learning is an intelligent solution to this problem, which uses a combination of dynamic programming and trial and error exploration to develop a set of optimal parameters for the Kalman filter [14]. In comparison to a typical iterative approach, it was found that using reinforcement learning led to slightly better estimates of the tuning parameter values; furthermore, the tuning process was performed in [14] with significantly less iteration, in comparison to an exhaustive search, due to the learning capability of the method. This benefits both the static parameters as well as the time varying parameters since the method is capable of constantly adapting the tuning based on collected navigation data. All these solutions, belonging to the adaptive Kalman filter approach, can reduce the uncertainty at the output of the Kalman filter but not completely. We present some results in Table 1.

**Table 1.** Performance comparison of different methods.

|  | [2] | [10] | [11] | [14] | **Proposed** |
|---|---|---|---|---|---|
| Accuracy | Prediction Error 2.97 m | RMS Error 0.29 m | Maximum Error $\leq 5$ m | Positional standard deviations 105 m | Estimation Error 0 m |
| Run time | 135 s | 20–30 min. | Not available | Iterative method | 30 s |

Let us consider now the solution based on post-processing the results of the Kalman filter. In this paper, we realize a treatment equivalent with the automatic tuning of the Kalman's filter hyper-parameters by the cooperation of the Hungarian data association algorithm with the Kalman filter. The Kalman filter is evaluated by the Hungarian algorithm. We do not need to use an adaptive Kalman filter to track the targets sensed by an automotive RADAR sensor, because the Hungarian data association algorithm "adapts" the data from the input of the Kalman filter. Hence, the cooperation between the Hungarian data association algorithm and the Kalman filter is equivalent with the building of an adaptive Kalman filter. The central part of our proposal consists of a post-processing step, based on polynomial fitting. This idea was exploited first in [16].

The algorithms in [2,10–14] can substantially reduce the uncertainty at the output of the Kalman filter but not completely. Contrary to the algorithms in [2,10–14], the explicit polynomial fitting algorithm and the wavelet-based polynomial fitting algorithm proposed

in this paper are able to completely reduce the uncertainty at the output of the Kalman filter, as is shown in the experiments reported in the previous section and is resumed in Table 1. So, the accuracy of targets' localization obtained by applying the proposed algorithms is higher than in the case of the algorithms proposed in [2,10–14]. The algorithms in [2,10–14] are used in different applications of the Kalman filter, comprising GPS, unmanned aerial vehicles (UAV), robots, automotive, navigation and tracking, implying some very complicated time trajectories. We concentrate in this paper only on automotive tracking. This choice explains the simple scenarios used in this paper, the targets' trajectories in the case of automotive RADAR being constrained by the physical characteristics of roads. By comparing the run time of the algorithms reported in [2,10,14] with the run time of the polynomial fitting algorithms proposed in this paper (see Table 1), we remark that our algorithms are faster. As future work, we intend to apply the proposed algorithms in more complex tracking scenarios. If it will work, we will try to implement the post-processing algorithms into a digital signal processor. One of the possible applications of target tracking is the classification of targets. Tacking as model the video tracking, we will try in the future to apply the classification method described in [26] for RADAR targets based on the proposed enhanced tracking algorithms.

## 6. Conclusions

We presented a new idea to reduce the uncertainty at the output of a Kalman filter or of a Kalman filter bank, used for target tracking in automotive RADAR sensor applications, by post-processing with polynomial fitting algorithms. Starting from the conjecture that all the trajectories of terrestrial maneuvering vehicles are polynomial in Cartesian coordinates, we have shown both theoretically and experimentally that a certain localization uncertainty appears at the output of a Kalman filter or of a Kalman filter bank. Based on previous research of the authors about single target tracking [16], and multiple target tracking [6], we have proposed here three polynomial fitting algorithms able to reduce the uncertainty at the output of a Kalman filter or a Kalman filter bank. Inspired by [6], we have highlighted, both theoretically and by simulation, the importance of the cooperation of the Kalman filter algorithm with the Hungarian data association algorithm in the case of multiple targets. We have shown that the uncertainty at the output of a Kalman filter or of a Kalman filter bank can be reduced, eventually losing the chronology, by applying polynomial fitting on both coordinate axes (implicit polynomial fitting). We have proposed three variants of data fitting: implicit [16], explicit and wavelet-based [6] polynomial fitting.

We have illustrated these methods via five simple numerical examples, realized using synthesized data (the first four experiments, the first two experiments are inspired from [6]) and real data (last experiment concerning a single target with linear trajectory, analyzed in [16] as well). We have shown that the explicit polynomial fitting algorithm is efficient in the case of multiple targets with linear trajectories, that the wavelet-based fitting algorithm is efficient in the case of a single target with non-linear trajectories and that the wavelet-based polynomial fitting and the explicit polynomial fitting outperform the implicit polynomial fitting algorithm. We proved their efficiency by comparison with adaptive Kalman filter solutions, as shown in Table 1.

**Author Contributions:** Conceptualization, G.M., R.L. and A.I.; methodology, G.M.; software, R.L.; validation, G.M., R.L. and A.I.; formal analysis, A.I.; investigation, G.M.; resources, A.I.; data curation, R.L.; writing—original draft preparation, G.M.; writing—review and editing, G.M.; visualization, R.L.; supervision, A.I.; project administration, G.M. All authors have read and agreed to the published version of the manuscript.

**Funding:** This research received no external funding.

**Institutional Review Board Statement:** Not applicable.

**Informed Consent Statement:** Not applicable.

**Data Availability Statement:** Not applicable.

**Acknowledgments:** We acknowledge the valuable comments of the reviewers whose remarks improved the quality of this paper. We have benefited from fruitful discussion with RADAR specialists, such as Hermann Rohling from Technical University Hamburg Harbor Germany and Ioan Nafornita from Politehnica University Timisoara Romania, who indicated some of the references used in this paper. We also want to thank Cristian Vesa and Cosmin Dună from Hella Timisoara for facilitating our access to the automotive RADAR sensor and to the data used in the last experiment reported in this paper.

**Conflicts of Interest:** The authors declare no conflict of interest.

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
