# Peer review of "Improving the Targets’ Trajectories Estimated by an Automotive RADAR Sensor Using Polynomial Fitting"

_applsci, doi:10.3390/app11010361_

Round 1
Reviewer 1 Report
The main weakness of the paper is that there is no experimental comparison with existing methods. The authors should give some experimental results obtained using competitive methods, described in section 5, and then outline adavantages of the presented approach.
Author Response
Response to reviewer 1
The main weakness of the paper is that there is no experimental comparison with existing methods. The authors should give some experimental results obtained using competitive methods, described in section 5,
We have included some comparisons in Table 1 from section 5 of the new submission concerning the accuracy and the speed of the proposed algorithms (between the lines 543 and 546). It is difficult to do these comparisons because different applications of Kalman filter implying different types of trajectories are considered in the papers mentioned in section 5.
and then outline adavantages of the presented approach.
We have outlined the advantages of the present approach at the end of section 5 of the new submission (between the lines 562 and 576) and at the end of section 6 (between the lines 597 and 600).

Reviewer 2 Report
The paper proposed three polynomial fitting algorithms to reduce the uncertainty of the Kalman filter's output used for targets tracking in automotive RADAR sensors applications. The results demonstrated the improvement of the estimated trajectory’s accuracy. I have a few comments below that I hope the authors could address in the revised manuscript:
- It looks to me that the paper is an extension of both the authors' conference papers [15] and [17]. Please indicate clearly in the revised manuscript what are new and different in this journal submission.
- In the Results section, the authors only showed the comparison among the proposed methods (i.e. polynomial fitting). I suggest the authors also compare with non-post-processing approaches and other post-processing approaches (if there is any).
- The authors mentioned once in the Discussion section that the proposed algorithms are faster than the algorithms in [2] and [13] but there were no run time numbers presented in the Results section. Please add more details of run time comparison in the Results section. I am curious about how the run times of the three proposed algorithms look like and how they are compared to other algorithms.
- The authors used the word "efficient" in the last paragraph of the Conclusions section. Do the authors mean "effective" or "efficient" or both?
- The authors used the word "hardly" in the Abstract and in the Introduction section. What does "hardly" mean here? Does it mean the accuracy of the proposed methods is the same as the other methods?
- It is not clear to me why a proposed fitting algorithm is better than other proposed fitting algorithms in a particular scenario but performs worse in another scenario. Please provide more insights when the authors compare the proposed methods in the Results section or in the Discussion section.
- I suggest move section 3.4 to section 4.
- There are a lot of typos, errors in grammar use, and word use throughout the paper. I suggest the authors revise the paper thoroughly. I am just listing below just to name a few:
- Please consider using commas instead of semicolons when listing. For example, in the Conclusions section, the sentence "We have proposed three variants of data fitting: implicit; explicit; and wavelets based; polynomial fitting" can be changed to "We have proposed three variants of data fitting: implicit, explicit, and wavelets based polynomial fitting". There are other places in the paper where semicolons were used. Please fix them too.
- Please consider merging short paragraphs into one paragraph in Section 2, section 3.2 and section 5. Using bullet points could be another option.
- This long sentence "We highlight, by simulations, the improvement of the estimated trajectory’s accuracy due to the polynomial fitting in case of single or multiple targets and we propose a new polynomial fitting method based on wavelets in two steps: denoising and polynomial part extraction, which compares favorably with classical polynomial fitting method" can be broken into two sentences.
- In the same sentence "the improvement of the estimated trajectory’s accuracy due to the polynomial fitting in case of single or multiple targets" can be changed to "the improvement of the estimated trajectory’s accuracy using the polynomial fitting for single and multiple targets".
- Also int the Abstract, "The effect of proposed post-processing methods is visible" should be "The effect of the proposed post-processing methods is visible"
- In the last paragraph of Section 3, "The polynomial fitting, as post processing method, was applied..." should be "The polynomial fitting, as a post processing method, was applied..."
- At the beginning of page 5, "Teoretically" should be "Theoretically".
- On page 7, "...means that we not rely very much..." should be "...means that we do not rely very much..."
- At the end of Section 3.2, "...the Kalman filter which traks a specified..." should be "...the Kalman filter which tracks a specified..."
- Please consider changing "conceive" to something else.
- Perhaps, please consider changing "Intermediary" to "Intermediate".
Author Response
- It looks to me that the paper is an extension of both the authors' conference papers [15] and [17]. Please indicate clearly in the revised manuscript what are new and different in this journal submission.
As we have written in the Cover Letter this submission is a proposal for a special issue of the international journal Applied Sciences entitled "Selected Papers from the 2020 43rd International Conference on Telecommunications and Signal Processing (TSP)". The submission extends the paper with ID: 32 and the title “Polynomial Based Kalman Filter Result Fitting to Data”, presented at the 2020 43rd International Conference on Telecommunications and Signal Processing (TSP) [15]. In the case of the conference paper, we have treated theoretically and by simulation only the case of single target tracking. Therefore, we have used the conference paper to write a part of section 3.2, dedicated to Kalman filters, a part of subsection 3.3.1 entitled Polynomial Fitting, subsection 3.3.2 - Wavelets Based Polynomial Fitting, the introduction of section 4, and a part of section 4.2 entitled Real Data. After the publication of [15], some of the authors of this submission have published a second conference paper, entitled “Multiple Radar Targets Tracking and Trajectories Fitting” in the Proceedings of International Symposium on Electronics and Telecommunications (ISETc), Timisoara, Romania, 05-06 November 2020 [17]. This paper treats only the case of multiple targets and we have used it to complete section 1-Introduction, section 2-Related Work, subsection 3.1 – Data association, and subsection 4.1-Simulated data, of the present submission.
We have expanded these conference papers to the size of a research article (20 pages); completing the first three sections with relevant details concerning our approach, especially subsection 3.1-Data association, adding new experimental results in subsection 4.1-Simulated Data and completing section 6-Conclusions. The section 5-Discussion of the submission is completely original.
We have completed the new variant of our submission with these remarks by updating the section 6-Conclusions, between the lines: 576-583.
- In the Results section, the authors only showed the comparison among the proposed methods (i.e. polynomial fitting). I suggest the authors also compare with non-post-processing approaches and other post-processing approaches (if there is any).We have included some comparisons in Table 1 from section 5 of the new submission, concerning the accuracy and the speed of the proposed algorithms (between the lines 534 and 536). It is difficult to make these comparisons because different applications of Kalman filter implying different types of trajectories (with different complexities) make the object of papers mentioned in section 5.
- The authors mentioned once in the Discussion section that the proposed algorithms are faster than the algorithms in [2] and [13] but there were no run time numbers presented in the Results section. Please add more details of run time comparison in the Results section.
We have mentioned that our algorithms are faster than the algorithm in [13] because the algorithm in [13] is iterative and our proposed algorithms are not iterative. An iterative algorithm supposes few iterations and each one takes a certain run time. We have mentioned the run time of the algorithm in [2] in Table 1.
I am curious about how the run times of the three proposed algorithms look like and how they are compared to other algorithms.
The run time of the proposed algorithms increases with the increasing of the input data volume (number of targets and number of positions of each target). The implicit polynomial fitting algorithm is faster than the explicit polynomial fitting algorithm. These algorithms are slower than the wavelets based fitting algorithm because they require two matrices inversions. The DWT algorithm is fast (even faster than the FFT algorithm sometimes). We have compared in Table 1 the run time of these algorithms with the run times of some algorithms found in literature, tacking as run time value for the proposed algorithms the maximal value obtained in the simulations presented in Section 4.
-
The authors used the word "efficient" in the last paragraph of the Conclusions section. Do the authors mean "effective" or "efficient" or both?
Efficient.
-
The authors used the word "hardly" in the Abstract and in the Introduction section. What does "hardly" mean here? Does it mean the accuracy of the proposed methods is the same as the other methods?It means that the accuracy of the proposed method is better than the accuracy of other methods.
-
It is not clear to me why a proposed fitting algorithm is better than other proposed fitting algorithms in a particular scenario but performs worse in another scenario.It was an error in the old submission, we have written implicit and the correct word is explicit. We have corrected this error in the new submission.
The run time of the proposed algorithms increases with the increasing of the input data volume (number of targets and number of positions of each target).
The traditional polynomial fitting algorithm operates with one-dimensional data. The results of the Kalman filtering represent the estimated coordinates of the targets and constitute a set of two-dimensional data. We proposed three algorithms: implicit, explicit, and wavelets based polynomial fitting, to solve the data-fitting problem in the case of two-dimensional data. Both implicit and explicit polynomial fitting algorithms suppose the separate polynomial fitting of data corresponding to each category of coordinates: horizontal and vertical. In the case of the implicit polynomial fitting algorithm, we obtain directly the result, by composing the two polynomials obtained previously. Therefore, the implicit polynomial fitting algorithm has three steps: traditional polynomial fitting applied to horizontal coordinates; traditional polynomial fitting applied to vertical coordinates; composition of the two polynomials obtained in previous steps.
However, the second one-dimensional data fitting procedure (which treats the vertical coordinates) could reject some data with extreme values, producing gaps in the result of the implicit polynomial fitting algorithm. We solved this drawback in the case of the explicit polynomial fitting algorithm by changing two steps of the implicit polynomial fitting algorithm: we invert the polynomial obtained after the separate polynomial fitting procedure applied to the horizontal coordinates and we apply the traditional one-dimensional fitting procedure for the vertical coordinates using as variable the result of the inversion. Therefore, the explicit polynomial fitting algorithm has the following three steps: traditional polynomial fitting applied to horizontal coordinates; inversion of the polynomial obtained; traditional polynomial fitting applied to vertical coordinates using as variable the result obtained at the end of the previous step. Hence, the accuracy of the explicit polynomial fitting procedure is better than the accuracy of the implicit polynomial fitting procedure. The run time of the explicit polynomial fitting algorithm is a little bit longer than the run time of the implicit polynomial fitting algorithm because the polynomial inversion step is more time demanding than the polynomial composition step.
The wavelet based fitting algorithm has the following five steps. The first step consists of denoising the data corresponding to horizontal coordinates. The aim of the second step is the wavelet-based polynomial fitting of the result obtained at the end of the previous step. We realize in the third step the denoising of data corresponding to vertical coordinates. The denoising conserves the extreme values of data. We realize in the fourth step the wavelet based polynomial fitting of the result obtained at the end of the previous step. The fifth step aims to realize the composition of the two polynomials obtained in the previous steps.
The accuracy of the wavelets based fitting algorithm is comparable with the accuracy of the explicit polynomial fitting algorithm because in both algorithms are conserved the extreme values of the vertical coordinates data. Despite the bigger number of steps, the run time of the wavelets based polynomial fitting algorithm is comparable with the run time of the implicit and explicit polynomial fitting algorithms because the DWT is a fast algorithm. - Please provide more insights when the authors compare the proposed methods in the Results section or in the Discussion section.
We have included some comparisons in Table 1 from section 5 of the new submission, concerning the accuracy and the speed of the proposed algorithms (between lines 534 and 536). It is difficult to make these comparisons because different applications of Kalman filter implying different types of trajectories (with different complexities) make the object of papers mentioned in section 5.
-
I suggest move section 3.4 to section 4.
We have moved section 3.4 to section 4 in the new submission.
-
There are a lot of typos, errors in grammar use, and word use throughout the paper. I suggest the authors revise the paper thoroughly.
We have read once again the entire paper and we have corrected some typos and errors in grammar use. We have divided some long sentences into two or three shorter sentences.
- I am just listing below just to name a few: Please consider using commas instead of semicolons when listing. For example, in the Conclusions section, the sentence "We have proposed three variants of data fitting: implicit; explicit; and wavelets based; polynomial fitting" can be changed to "We have proposed three variants of data fitting: implicit, explicit, and wavelets based polynomial fitting".
We have operated this modification in the new variant of submission.
- There are other places in the paper where semicolons were used. Please fix them too.
We have made the same type of modification a few times in the new submission. We have also erased some repetitions.
- Please consider merging short paragraphs into one paragraph in Section 2, section 3.2 and section 5. Using bullet points could be another option.
We have merged short paragraphs in four paragraphs in section 2, in a single paragraph in section 3.2, and four paragraphs in section 5.
- This long sentence "We highlight, by simulations, the improvement of the estimated trajectory’s accuracy due to the polynomial fitting in case of single or multiple targets and we propose a new polynomial fitting method based on wavelets in two steps: denoising and polynomial part extraction, which compares favorably with classical polynomial fitting method" can be broken into two sentences.
We have divided this long sentence into two shorter sentences in the new submission.
- In the same sentence "the improvement of the estimated trajectory’s accuracy due to the polynomial fitting in case of single or multiple targets" can be changed to "the improvement of the estimated trajectory’s accuracy using the polynomial fitting for single and multiple targets".
We have made this change in the new submission.
- Also int the Abstract, "The effect of proposed post-processing methods is visible" should be "The effect of the proposed post-processing methods is visible".
We have made this correction in the new submission.
- In the last paragraph of Section 3, "The polynomial fitting, as post processing method, was applied..." should be "The polynomial fitting, as a post processing method, was applied...".
We have made this modification at the end of section 2 in the new submission.
- At the beginning of page 5, "Teoretically" should be "Theoretically".
We have made this modification on page 5 of the new submission.
- On page 7, "...means that we not rely very much..." should be "...means that we do not rely very much...".
We have made this modification on page 7 of the new submission.
- At the end of Section 3.2, "...the Kalman filter which traks a specified..." should be "...the Kalman filter which tracks a specified..."
We have made this modification at the end of section 3.2 in the new submission.
- Please consider changing "conceive" to something else.
We have replaced five times the word conceived with the word designed and one time with the word realized in the new submission.
- Perhaps, please consider changing "Intermediary" to "Intermediate".
We have replaced three times the word Intermediary with the word Partial in the captions of Figures 3, 4 and 5 of the new submission.
Please receive a warm Thank You for your help in improving the quality of our submission.

Reviewer 3 Report
REVIEW
Article titled: “Improving the Targets’ Trajectories Estimated by an Automotive RADAR Sensor Using Polynomial Fitting”
Applied Sciences no. 1033498
List of Authors:
Georgiana Magu, Radu Lucaciu, Alexandru Isar
- In this paper the Authors present the problem of targets tracking using the measurements of an automotive RADAR sensor based on polynomial fitting. As it is commonly known, multiple target tracking is realized by the cooperation of two algorithms, i.e. a measurement-to-track data association algorithm and a tracks filtering algorithm. Usually, tracks filtering is realized by Kalman filters. Kalman filters are widely used to estimate the state of a linear dynamical system from noisy measurements. Despite its wide use, the Kalman filter is not completely automated yet.
Taking the above into account, the Authors developed a different semi-blind post processing approach, which is faster and more robust. Starting from the conjecture that the trajectory is polynomial in Cartesian coordinates, proposed method supposes to fit the data obtained at the output of the tracker to a polynomial, which seems to be an interesting approach.
- In introduction (Section titled: Related Work), the Authors made extensive literature review dealing with the most important aspects of targets tracking techniques, Kalman filtering problems, and more recent approaches for the reduction of the uncertainty of Kalman filters employ different optimization approaches, including the simplex algorithm and machine learning algorithms, statistical consistency tests, and many others. Tuning Kalman filters is difficult and often is done before navigation by the filter designer. This process is very time consuming and no way guaranteed to result in optimal parameters.
Bearing in mind a more efficient optimization algorithm and the process of “corresponding frames of video recordings grouped”, the following article concerning the similar problem entitled: A geometrical divide of data particle in gravitational classification of Moons and Circles data sets, Entropy 2020, 22(10), is also supposed to be listed in the References.
Other comments:
- In Chapter 3, Section 3.2 (titled: Kalman Filter) the equation (1) should be verified: in my opinion symbol ‘ denotes the transpose - should be replaced by another transposition mark, for example “ T ”.
What does the letter T in equations No. (6) mean?
Section 3.3.1., entitled: Polynomial Fitting, line 244: the Authors wrote:
of a set of data yt, t=1…M.
It should be written: of a set of data yt, t=1,…,M
Line 254-255: The disadvantage of this data fitting method is the necessity to a priori know the degree of the polynomial f, n. The notation is incomprehensible.
Conclusion. The Authors should carefully review all the mathematical formulas in terms of their correctness.
Figure no. 4 is too small in contrast to Figure no. 5. For this reason, Figure no. 4 is difficult to read. It should be enlarged.
Figure no. 6 is not correct. The axis of this drawing is missing.
- What were the specific parameters of the RADAR sensor that was used in this experiment?
- In Conclusion (line 577-582) the Authors wrote: “…We have shown that the explicit polynomial fitting algorithm is efficient in case of multiple targets with linear trajectories, that the wavelets based fitting algorithm is efficient in case of single target with non-linear trajectories and that the wavelets based polynomial fitting outperforms the implicit polynomial fitting algorithm..”
My question is:
In what specific way did the authors demonstrate the effectiveness of the developed algorithm?
- There is also no comment from the Authors on what the computational burden of the proposed method is and if their solution is used in equipment working in real conditions
The Authors addressed a problem which is relevant and appealing for this journal. However, I cannot recommend the current manuscript for publication unless the current version is corrected. After providing the amendments to the article, the work ought to be reviewed once again.

Author Response
Response to Reviewer 2
...Tuning Kalman filters is difficult and often is done before navigation by the filter designer. This process is very time consuming and no way guaranteed to result in optimal parameters.
Bearing in mind a more efficient optimization algorithm and the process of “corresponding frames of video recordings grouped”, the following article concerning the similar problem entitled: A geometrical divide of data particle in gravitational classification of Moons and Circles data sets, Entropy 2020, 22(10), is also supposed to be listed in the References.
We have read this excelent article and we think that it represent a fine contribution in the field of video tracking but the objective of our proposal is limited to automotive RADAR tracking.
Other comments:
In Chapter 3, Section 3.2 (titled: Kalman Filter) the equation (1) should be verified: in my opinion symbol ‘ denotes the transpose - should be replaced by another transposition mark, for example “ T ”.
We have made this modification, see equation (1) and the line 198 in the new submission.
What does the letter T in equations No. (6) mean?
The transpose.
Section 3.3.1., entitled: Polynomial Fitting, line 244: the Authors wrote:
of a set of data yt, t=1…M.
It should be written: of a set of data yt, t=1,…,M
We have made this modification in the new submission.
Line 254-255: The disadvantage of this data fitting method is the necessity to a priori know the degree of the polynomial f, n. The notation is incomprehensible.
We have corrected this notation using the notation in equation (12) in the new submission.
Conclusion. The Authors should carefully review all the mathematical formulas in terms of their correctness.
We have reviewed the formulas in the new submission.
Figure no. 4 is too small in contrast to Figure no. 5. For this reason, Figure no. 4 is difficult to read. It should be enlarged.
We have enlarged Figure no. 4 in the new submission.
Figure no. 6 is not correct. The axis of this drawing is missing.
We have completed the axis in Figure no. 6 in the new submission.
What were the specific parameters of the RADAR sensor that was used in this experiment?
We have specified the parameters of the RADAR sensor used in the experiment with real data in the new submission (between the lines 446 and 458).
In Conclusion (line 577-582) the Authors wrote: “…We have shown that the explicit polynomial fitting algorithm is efficient in case of multiple targets with linear trajectories, that the wavelets based fitting algorithm is efficient in case of single target with non-linear trajectories and that the wavelets based polynomial fitting outperforms the implicit polynomial fitting algorithm..”
My question is:
In what specific way did the authors demonstrate the effectiveness of the developed algorithm?
We have demonstrate the effectiveness of the proposed algorithm by comparing the accuracy of the trajectory estimation obtained with the accuracies obtained applying other trajectory’s correction methods (see the Discussion section, especially the lines 544-547 and the lines 562-569), in the new submission. We also compared the run time of these algorithms in the new submission (between lines 574 and 576).
There is also no comment from the Authors on what the computational burden of the proposed method is
We also compared the run time of these algorithms in the new submission (between lines 574 and 576).
and if their solution is used in equipment working in real conditions.
The solution could be used in equipment working in real conditions but it is not used yet. Please read the lines 577 to 579 in the new submission.
The Authors addressed a problem which is relevant and appealing for this journal. However, I cannot recommend the current manuscript for publication unless the current version is corrected. After providing the amendments to the article, the work ought to be reviewed once again.
Thank you for the valuable comments.

Round 2
Reviewer 2 Report
The authors addressed my comments in the revised manuscript and in the cover letter.
Author Response
Thank you very much.
With consideration!
Reviewer 3 Report
REVIEW_2
Article titled: “Improving the Targets’ Trajectories Estimated by an Automotive RADAR Sensor Using Polynomial Fitting”
Applied Sciences no. 1033498
List of Authors:
Georgiana Magu, Radu Lucaciu, Alexandru Isar
The article Applied Sciences no. 1033498 entitled “Improving the Targets’ Trajectories Estimated by an Automotive RADAR Sensor Using Polynomial Fitting” has been carefully modified and well revised.
However, there are some mistakes that requires absolute improvement.
- According to my previous comments - the following article titled: “A geometrical divide of data particle in gravitational classification of Moons and Circles data sets, Entropy 2020, 22(10)” - should be listed in the References.
In the revised version and Response to Reviewer Comments – the Authors wrote: “…We have read this excellent article and we think that it represent a fine contribution in the field of video tracking but the objective of our proposal is limited to automotive RADAR tracking …”
I believe that the mechanisms included in the article entitled “A geometrical divide of data particle in gravitational classification of Moons and Circles data sets, Entropy 2020, 22 (10), can be successfully applied in the “automotive radar tracking” process. For this reason, I believe that a suggested article should be included in the reference list of a peer-reviewed article.
- There is also no comment from the Authors on what the computational burden of the proposed method is and if their solution is used in equipment working in real conditions.
In the revised version and Response to Reviewer Comments – the Authors wrote: “… We also compared the run time of these algorithms in the new submission (between lines 574 and 576 - This is not the answer to my comment.
My comment concerned the computational effort of the proposed algorithm/method. Please modify your answer.
The work is supposed to be finally accepted for publication in Applied Sciences, after providing the amendments to the article.

Author Response
I used black for review comments and red to respond to comments.
- According to my previous comments - the following article A geometrical divide of data particle in gravitational classification of Moons and Circles data sets, Entropy - should be listed in the References.
In the revised version and Response to Reviewer Comments the Authors wrote: We have read this excellent article and we think that it represent a fine contribution in the field of video tracking but the objective of our proposal is limited to automotive RADAR tracking.
The article A geometrical divide of data particle in gravitational classification of Moons and Circles data sets, Entropy 2020, 22 (10), can be successfully applied in the automotive radar tracking process.
For this reason, I believe that a suggested article should be included in the reference list of a peer reviewed article.
The article “A geometrical divide of data particle in the gravitational classification of Moons and Circles data sets”, Entropy 2020, 22, (10), deals with image classification. Our proposal deals with targets’ tracking. Targets’ tracking is a processing method applied before classification. We intend to use the proposed algorithms for targets’ classification in the future and we will tray the algorithm described in the article “A geometrical divide of data particle in the gravitational classification of Moons and Circles data sets”, Entropy 2020, 22 (10), [26], as we have mentioned in the new submission between lines 656 and 658.
- There is also no comment from the Authors on what the computational burden of the proposed method is and if their solution is used in equipment working in real conditions.
In the revised version and Response to Reviewer Comments the Authors wrote:
We also compared the run time of these algorithms in the new submission (between lines 574 and 576 - This is not the answer to my comment. My comment concerned the computational effort of the proposed algorithm/method.
Please modify your answer.
Our algorithms are simpler than the algorithms in literature we compared with. This is the reason why the computational burden of the implementation of the proposed algorithms is smaller than the computational burden of the algorithms reported in the literature, used for comparison in our submission. A measure for the computational burden of an algorithm is the run time of that algorithm. For this reason, we have responded to your question that we have compared favorably the run time of our algorithms with the run time of the algorithms found in the literature. In what concerns the utilization of the proposed algorithms in equipment working in real conditions, in our knowledge, there are no other articles in literature proposing the polynomial fitting as a method to improve the accuracy of the trajectories’ estimation with Kalman filters or Kalman filters banks. In consequence, there is not designed yet equipment using the solution proposed. However, we could design such equipment, as we mention in the new submission between lines 654 and 655. Concerning the functioning in real-time, we believe that the proposed algorithms fulfill this condition, taking into account the fact that we have synchronized the movie that represents the input video data in our last experiment with the tracking obtained using the wavelets based polynomial fitting method proposed.
